# What Makes a Good Explanation?: A Harmonized View of Properties of Explanations

**Zixi Chen**[*], **Varshini Subhash**[*], **Marton Havasi, Weiwei Pan, Finale Doshi-Velez**
John A. Paulson School of Engineering and Applied Sciences
Harvard University
{zixichen, varshinisubhash, mhavasi, weiweipan}@g.harvard.edu
finale@seas.harvard.edu

## Abstract

Interpretability provides a means for humans to verify aspects of machine learning (ML) models. Different tasks require explanations with different properties. However, presently, there is a lack of standardization in assessing properties of explanations: different papers use the same term to mean different quantities, and different terms to mean the same quantity. This lack of standardization prevents us from rigorously comparing explanation systems. In this work, we survey explanation properties defined in the current interpretable ML literature, we synthesize properties based on what they measure, and describe the trade-offs between different formulations of these properties. We provide a unifying framework for comparing properties of interpretable ML.

## 1 Introduction

Interest in interpretable machine learning[2] has grown in recent years. It is accepted that different contexts will require different interpretable ML methods. For example, the kind of explanation required to determine if an early cardiac arrest warning system is ready to be integrated into a care setting is very different from the type of explanation required for a loan applicant to help determine the actions they might need to take to make their application successful. Identifying which interpretability methods are best suited to which tasks remains a grand challenge in interpretable ML.

We can quantify how "good" an interpretable machine learning method is by quantifying the *properties* it preserves. Since these properties must be tied to the task it is designed to explain, they can serve as an abstraction between the method and its context. For example, both an explanation describing features a patient might change to reduce future cardiac risk and an explanation describing features a loan applicant might change to successfully get a loan, should clearly have no false positives—if the person acts on the listed features, the output should change. In both cases, it might also be important for the explanation to be reasonably short as the person may not have the time or inclination to parse through a complex description. In contrast, if one were vetting a risk predictor for a health or justice scenario, one might require a more complete explanation of the entire model—and not mind spending time carefully inspecting it. In other words, desirable properties in explanations are completely dependent on the context and the explanation being used, which is why we can consider them as an abstraction between the two. If one knows what properties are needed for what contexts, then one might be able to check for those properties computationally to identify promising interpretable machine learning methods prior to more expensive user studies with people.

---

[*]Equal Contribution

[2]In this paper, we will use the term interpretable machine learning synonymously with explainable AI.

2022 Trustworthy and Socially Responsible Machine Learning (TSRML 2022) co-located with NeurIPS 2022.

Unfortunately, while many works have defined properties, there is little current consensus around the terminology and formulation of interpretable machine learning properties. We highlight some of the issues due to this. First, different works have used different terms for the same property. For example, one work might call a property robustness while another calls it stability. Second, different works have formalized the notion of these properties differently: the expressions that one work uses to computationally evaluate compactness may be different than another. The current state of having multiple definitions in literature and a lack of consistent formulations makes it difficult to compare methods rigorously. It also becomes challenging to interpret what it truly means when one claims that a certain context needs a certain property or that a certain algorithm preserves it.

This paper reviews and synthesizes existing properties and definitions in the interpretable ML literature. We first collect metrics describing the same properties but termed differently in different works. For each property, we (a) precisely describe the various mathematical formulations that have been proposed, (b) identify the key variations in those formulations, and (c) describe how different formulations of the same property might be appropriate in different contexts. Finally, we use these formulations to make precise how different properties may or may not be in tension with each other.

Our framework provides a much-needed systematic synthesis of a key part of the interpretable machine learning ecosystem. Our work serves as a reference for not only common properties we want from interpretable machine learning methods, but also what considerations inform how the abstract property is made precise. Our work can serve as a reference for both what terms are used to describe properties as well as how to formalize them for future research in interpretable machine learning.

**Notation and Terminology**  In this work, we use $f$ to denote the model to be explained and $E(f)$ to represent the explanation; we define the *explanation* as the information provided from the model to the user. Throughout the work, we look at predictive models that yield a prediction $\hat{y} = f(\mathbf{x})$ for the input point $\mathbf{x}$ that has $K$ features $x^{(1)} \ldots x^{(K)}$. For the model $f$, we denote its prediction as $\hat{y}_f$, which may be discrete or continuous depending on whether the task is classification or regression. If the explanation depends also on the input, we will use the notation $E(f, \mathbf{x})$. Certain evaluations of explanations also require a baseline or reference input value, we denote $\mathbf{x}_0$ as this reference value. Some take into account the ground truth value at an input $\mathbf{x}$, and we denote $y$ as the respective ground truth. For feature attribution explanations, the *ranking* of features $\text{Rank}_E(f, \mathbf{x})$ given the attribution weights $E(f, \mathbf{x})_k$ is the feature ordering from largest to smallest. We use $\ell(\cdot, \cdot)$ to denote norms and specify the type of norm (e.g. $\ell_2$ for the $L$-2 norm) in the text. When comparing the model prediction output, $\hat{y}_f = f(\mathbf{x})$, to the output implied by the explanation, $\hat{y}_E$ (where $\hat{y}_E = E(f, \mathbf{x})(\mathbf{x})$ for local explanations and $\hat{y}_E = E(f)(\mathbf{x})$ for global ones), we use $\mathcal{L}(\hat{y}_f, \hat{y}_E)$ to denote the loss that describes how well the explanation captures the model's behavior at input $\mathbf{x}$.

## 2   Framework for Synthesizing Properties of Model Explanation

We synthesize explanation evaluation metrics from existing literature and find that they broadly fall under four distinct categories – *sensitivity*, *faithfulness*, *complexity* and *homogeneity*, each of which corresponds to a desired property for model explanations. Note that while the metrics can be model-specific, the properties are independent of the type of explanations. We provide a visual categorization of the metrics introduced by various works in literature in Figure 1.

Given page limitations, we provide only a partial synthesis of the properties of sensitivity and faithfulness with respect to the various explanation metrics proposed in literature. We categorize these metrics by their mathematical formulation, the underlying notion being captured and the human tasks requiring the preservation of these properties. A complete synthesis is at `https://arxiv.org/abs/2211.05667`.

### 2.1   Robustness and Sensitivity

We first examine *robustness*, also often referred-to as *sensitivity*. For local interpretability methods (i.e. methods that explain the prediction for a given $\mathbf{x}$), sensitivity measures the similarity of explanations under changes to the input point $\mathbf{x}$. It has been shown that robustness increases user trust in the explanation (Yeh et al., 2019a; Ghorbani et al., 2017): users expect explanations to be stable under minor changes to the input point $\mathbf{x}$. In high-stakes applications, it is important to ensure that minor changes to the input do not lead to drastically different explanations since this could mislead users.

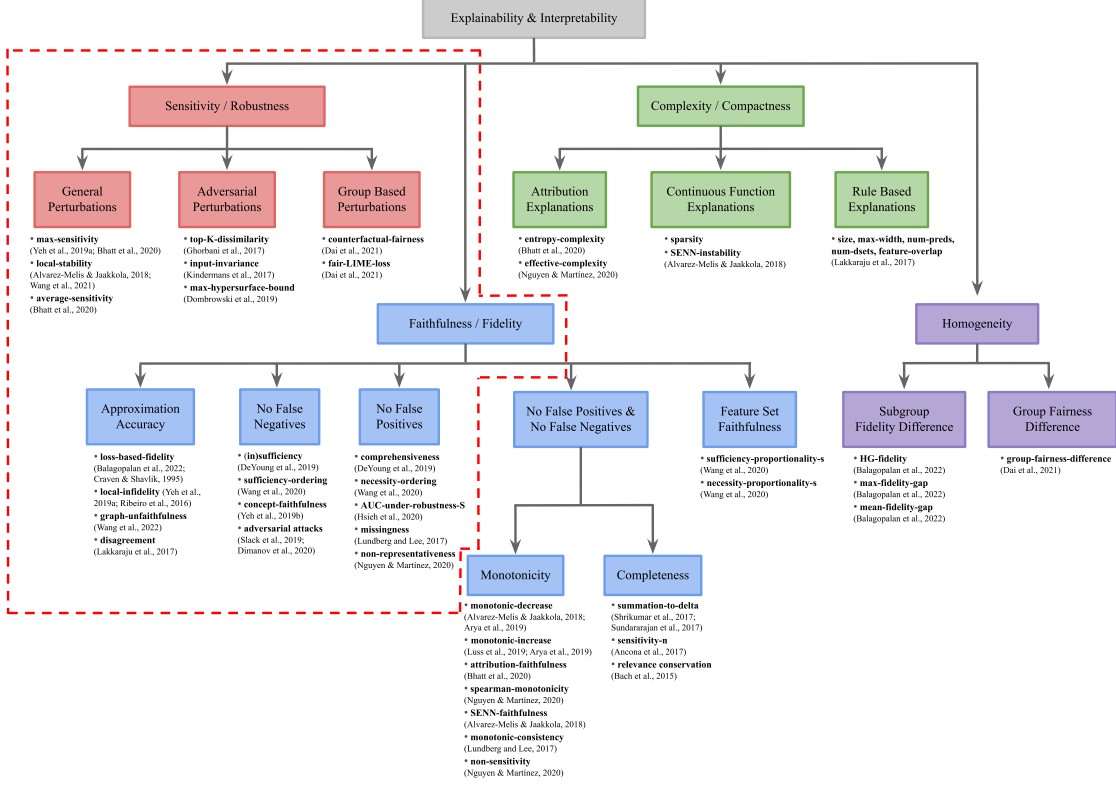

Figure 1: **Property Synthesis Framework** – Various metrics in literature fall under four main properties – *sensitivity*, *faithfulness*, *complexity* and *homogeneity*. Further categorization groups these metrics based on similarities in mathematical formulation. We discuss those enclosed in the dotted red box in this paper and refer the reader to the full-length paper for the complete discussion.

Additionally, sensitivity to input perturbations plays a key role in determining the success of an adversarial attack that seeks to perturb the explanation via an imperceptible perturbation of the input.

In the following, we categorize and compare proposed mathematical formulations for sensitivity. In all cases, we presume there is some context-dependent definition of similarity between two explanations.

### 2.1.1 Sensitivity via General Perturbations

General perturbations usually refer to random perturbations within a specific region around the original input and the examples in literature use spherical regions for local perturbations and cylindrical regions for group-based perturbations. However, a general definition of sensitivity could use geometric regions of different shapes, depending on the application.

The first metric of sensitivity via general perturbations, proposed by Yeh et al. (2019a) (also discussed by Bhatt et al. (2020)), defines **max-sensitivity** as the maximum change in the explanation $E$ under a small perturbation. The perturbation is defined within a sphere of radius $r$ around the input $\mathbf{x}$ and the change in explanation is measured by the $\ell_2$ norm:

$$\text{max-sensitivity}(E, f, \mathbf{x}, r) \triangleq \max_{\|\mathbf{x}'-\mathbf{x}\| \leq r} \|E(f, \mathbf{x}') - E(f, \mathbf{x})\| \tag{1}$$

While bounding max-sensitivity bounds the change in explanation within a small region, it does not require continuity. That is, the explanation may change abruptly within the region. In contrast, Alvarez-Melis and Jaakkola (2018) (also discussed in Yeh et al. (2019a)) propose an alternative metric, called **local-stability**, that measures the ratio of the maximum change in explanation within

the perturbation region to the size of the perturbation region (akin to Lipschitz continuity). Again, the change in the explanation $E$ and the distance from $\mathbf{x}$ are both measured by the $\ell_2$ norm:

$$\text{local-stability}(E, f, \mathbf{x}, r) \triangleq \max_{\|\mathbf{x}'-\mathbf{x}\| \leq r} \frac{\|E(f, \mathbf{x}) - E(f, \mathbf{x}')\|}{\|\mathbf{x} - \mathbf{x}'\|} \tag{2}$$

Wang et al. (2021) refer to robustness for a feature attribution $E$ as a constrained version of Equation 2, wherein an explanation is **locally-robust** if its local-stability$(E, f, \mathbf{x}, r) \leq \lambda$ for a given threshold $\lambda$.

Unlike max-sensitivity, bounding local-stability ensures that the perturbed input explanations become sufficiently similar to the explanation at input $\mathbf{x}$, as we approach $\mathbf{x}$. In fact, as $r \to 0$, local-stability converges to the $\ell_2$ norm of the gradient of $f$ at $\mathbf{x}$, and can be approximated for small $r$ (if $f$ is differentiable). However, bounding local-stability means is that explanations become insensitive to all small input perturbations, including the ones that cause significant changes to model prediction.

In cases where the explanation varies smoothly in a region of perturbation except for at a few isolated points, max-sensitivity or local-stability may be unrepresentative of the behavior of the explanation system, as it can be determined completely by these "outlier points". To address this, we can consider **average-sensitivity** (Bhatt et al., 2020), by looking at the average change in the explanation within the perturbation region. To take the average, they define a distribution $p(\mathbf{x})$, which captures the region of interest. In the simplest formulation, $p(\mathbf{x})$ is uniformly distributed in a sphere with radius $r$ around $\mathbf{x}$: $p = U(\{\mathbf{x}' | \|\mathbf{x}' - \mathbf{x}\| \leq r\})$. To measure the change in the explanation we use $\ell_2$ distance, although in the original formulation the metric is stated with a general distance metric.

$$\text{average-sensitivity}(E, f, \mathbf{x}, p) \triangleq \int_{\mathbf{x}' \in \mathrm{R}^K} \|E(f, \mathbf{x}) - E(f, \mathbf{x}')\| \, p(\mathbf{x}') \, \mathrm{d}\mathbf{x}' \tag{3}$$

An additional benefit of Definition 3 is that one can obtain an unbiased estimate by drawing Monte-Carlo samples from $p(\mathbf{x})$. The same does not hold for max-sensitivity and local-stability. Further, explanation aggregation can be used as a means for minimizing average-sensitivity.

### 2.1.2 Sensitivity via Adversarial Perturbations

While general perturbations usually refer to random perturbations in a geometric region surrounding the original input, adversarial perturbations refer to well-designed perturbations on some targeted features in a specific direction such that the change in model behavior is infinitesimal while the explanation undergoes considerable change, i.e. is highly sensitive. The goal with adversarial perturbations is to find the worst perturbation that causes the explanation to behave sensitively, which resembles the idea of maximum sensitivity described in Equation 1.

(Ghorbani et al., 2017) denote sensitivity as fragility of neural network explanations, formally computed with dissimilarity metrics measuring explanation change. The inputs are images, explanations are feature importance maps or influence functions and each dissimilarity metric corresponds to a region of perturbation $R = x' : x + c$ for all $c$ in the input. Each dissimilarity metric measures the explanation change obtained by iteratively optimizing towards the worst-case adversarial perturbation.

The first dissimilarity metric **top-K-dissimilarity** is defined as the minimization of the sum of importances of top $K$ initially most important features, where $E(f, \mathbf{x})_k$ denotes the $k^{th}$ important feature in an explanation, and $\mathbf{x}'_{\text{topK}}$ is the input $\mathbf{x}$ where only the top $K$ features are perturbed:

$$\text{top-K-dissimilarity}(E, f, \mathbf{x}) \triangleq \max_{\|\mathbf{x}'_{\text{topK}} - \mathbf{x}\| \leq r} \left[ -\sum_{k=1}^{K} E(f, \mathbf{x}'_{\text{topK}})_k \right] \tag{4}$$

This can also be viewed as maximization over the set of all possible input perturbations, such that the original $K$ top features are perturbed to have minimal importance scores. This is the same as finding a perturbed explanation that is farthest from the unperturbed explanation as in max-sensitivity (Equation 1), with the difference that the region of perturbation $R$ is defined as the top $K$ most important features but not all features. *Remaining metrics are analyzed in our full paper version.*

(Kindermans et al., 2017) propose **input-invariance** as a necessary condition for ensuring insensitivity of saliency based feature attribution methods. This requires that a constant shift in the input $\mathbf{x}$, which

does not affect model predictions or weights, must not affect the explanation (attribution) either. The sensitivity is measured by experimentally comparing the saliency heatmaps of the unperturbed and perturbed inputs, which should be identical if input-invariance is satisfied.

As discussed in Section 2.1.1, the metric local-stability (Equation 2) by (Alvarez-Melis and Jaakkola, 2018) has the drawback of being insensitive even to those adversarial perturbations which cause change in model output. Explanations which are insensitive to such attacks on the model can be misleading and lead to dangerous downstream consequences. To address this, (Dombrowski et al., 2019) **bound the curvature of the hypersurface** of a constant network output manifold $M = \{\mathbf{x} \in \mathbb{R}^d | f(\mathbf{x}) = c\}$ for a constant $c$. Adversarial perturbations will then lie on this manifold. The larger the curvature of this hypersurface, the higher the sensitivity of the saliency map explanation.

### 2.1.3  Sensitivity via Group Based Perturbations

Group-based perturbations can be defined as changing the group membership of one of the features of the input $\mathbf{x}$. In contrast to a spherical radius of perturbation, this corresponds to a cylindrical region in which the "non-sensitive" features are fixed and the "sensitive" features can vary over their full range. Due to its implications on fairness, it becomes important to study the sensitivity of explanations subject to group perturbations along different regions of the cylinder.

For instance, if we have gender as a feature with group values contained in the set {*male*, *female*, *other*}, then a group-based perturbation would change an input point's group membership from one value to another. This kind of perturbation is very relevant to checking for and maintaining fairness across groups. Sensitivity via such group-based perturbations checks for the change in the explanation, when the only difference between the original input $\mathbf{x}$ and the perturbed input $\mathbf{x}'$ is their demographic group membership. Fairness preserving explanations must capture the underlying behavior of the model accurately and faithfully, which means that a fairness preserving model must have an explanation that reflects this fairness, while an unfair model must have an explanation revealing the unfairness. This can be formalized as a condition for preserving **counterfactual-fairness**, by Dai et al. (2021), where the change in explanation must be approximately equal to the change in model output, when the input is subjected to a group-based perturbation:

$$E(f, \mathbf{x}) - E(f, \mathbf{x}') \approx f(\mathbf{x}) - f(\mathbf{x}') \tag{5}$$

As an example of practical usage, Dai et al. (2021) consider LIME and propose adding a penalty term for fairness to the optimization objective to generate fairness-preserving explanations. $\pi_{\mathbf{x}}$ is a distance metric defining the local neighborhood of an input, $\lambda_1$ is the tuning parameter for complexity $\Omega$ and $\lambda_2$ is the tuning parameter for the fairness-preservation term $\psi$.

$$\text{fair-LIME-loss}(E, f, \mathbf{x}) \triangleq \mathcal{L}\left(E, f, \pi_{\mathbf{x}}\right) + \lambda_1 \Omega(E, \mathbf{x}) + \lambda_2 \psi(f, E) \tag{6}$$

## 2.2  Faithfulness and Fidelity

In the literature, *fidelity* and *faithfulness* are often used interchangeably to describe the ability of the explanation to capture the true underlying behavior of the model. Faithfulness is desirable because a good explanation aims to reveal the true reasoning and decision making of a complex model to the user. If an explanation does not explain the model's behavior accurately (faithfully), it could mislead users into using the model's decisions and predictions for high-stakes situations and lead to undesirable consequences and low user trust. While the notion of the explanation being true to the underlying model is intuitive, it can be formalized in many different ways. We describe a few here, and expand in our full paper version `https://arxiv.org/abs/2211.05667`.

### 2.2.1  Faithfulness as Approximation Accuracy

Faithfulness as *approximation accuracy* measures how well an explanation approximates the model's behavior overall. This can be described using a loss to measure how similar the outputs implied by the explanation(s) are to the model outputs.

Balagopalan et al. (2022) adopt a general definition called **loss-based-fidelity** from Craven and Shavlik (1995) and average the quality of approximation around all inputs $\mathbf{x} \in \mathcal{X}$ as:

$$\text{loss-based-fidelity}(E, f) \triangleq \frac{1}{|\mathcal{X}|} \sum_{\mathbf{x} \in \mathcal{X}} \mathcal{L}\Big(f(\mathbf{x}), E(f)(\mathbf{x})\Big) \tag{7}$$

The choice for the loss $\mathcal{L}$ can be an appropriate task-specific metric such as accuracy, AUROC or mean error. Note that this formulation measures the faithfulness of a global explanation at all inputs; if the explanations are local, then we can modify it to measure the average degree of faithfulness of each local explanation at each input $\mathbf{x}$. Lundberg and Lee (2017) also refer to faithfulness for a local explanation at $\mathbf{x}$ as a constrained version of Equation 7, wherin an explanation satisfies **local-accuracy** if the inner loss term is simple difference and has a value of 0, i.e. $\mathcal{L}\big(f(\mathbf{x}), E(f, \mathbf{x})(\mathbf{x})\big) = f(\mathbf{x}) - E(f, \mathbf{x})(\mathbf{x}) = 0$.

A specific instantiation of Equation 7 can be seen in Yeh et al. (2019a), where the loss is given by the $\ell_2$ distance between the change in model output and change in explanation output (squared loss) when the input is perturbed: $\sum_{\mathbf{x}'} \ell_2(f(\mathbf{x}') - f(\mathbf{x}), \hat{y}_E(\mathbf{x}') - \hat{y}_E(\mathbf{x}))$. (Recall that $\hat{y}_E$ is the predicted value of $y$ given by the explanation). This looks at the local quality of the approximation around an input $\mathbf{x}$. Yeh et al. (2019a) call this **local-infidelity**; if the explanation is a linear approximation to the model at $\mathbf{x}$, this becomes:

$$\text{local-infidelity}(E, f, \mathbf{x}, p) \triangleq \mathbb{E}_{p(\mathbf{x}')} \left[ \left( (\mathbf{x}' - \mathbf{x})^T E(f, \mathbf{x}) - (f(\mathbf{x}) - f(\mathbf{x}')) \right)^2 \right] \tag{8}$$

Here, $\mathbf{x}'$ are drawn from some $p(\mathbf{x}')$ centered at the input of interest $\mathbf{x}$ and the explanation $E(f, \mathbf{x})$ is the set of linear approximation weights. This approach is also used to create explanations in Ribeiro et al. (2016) (LIME): LIME finds the set of weights with least error according to Equation 8. The primary difference between Equations 7 and 8 is that local-infidelity samples around a local input $\mathbf{x}$, unlike loss-based-fidelity which considers all inputs.

### 2.2.2 Faithfulness via No False Negatives

Faithfulness as *no false negatives* evaluates the degree to which nothing important is left out in the explanation. While this notion of faithfulness is most applicable to feature attribution methods—e.g. we do not want important features left out—no false negatives has also been applied to explanations derived from concept models.

DeYoung et al. (2019) consider feature attribution methods that assign non-negative importance scores to the $K$ features, with higher values indicating a more important feature for the model's function approximation $f$. They capture the notion of faithfulness as a ranking of the features: retaining the most important features should be sufficient for computing the value(s) of $f$. Let $\mathbf{x}_{E_s}$ denote the input $\mathbf{x}$ such that the *retention proportion* $s \in [0, 1]$ is the proportion of the most important features retained. For features indexed $k = 1 \ldots K$, the number of retained features will be $\lceil sK \rceil$. $\text{Rank}_E(f, \mathbf{x})$ is the *ranking* of the $K$ features from highest to lowest and $\mathbf{x}_0$ is the reference value for each feature:

$$\mathbf{x}_{E_s}^{(k)} = \mathbf{x}^{(k)} \text{ if } k \in \{\text{Rank}_E(f, \mathbf{x})_1 \ldots \text{Rank}_E(f, \mathbf{x})_{\lceil sK \rceil}\} \text{ else } \mathbf{x}_0^{(k)} \tag{9}$$

The metric **(in)sufficiency** is measured as a function of the portion of retained important features $s$:

$$\text{(in)sufficiency}(f, \mathbf{x}, E, s) \triangleq |f(\mathbf{x}) - f(\mathbf{x}_{Es})| \tag{10}$$

Note that the original formulation omits taking the absolute value, as it only considers the confidence level in a classification task. Here we present a generalized version that applies to all $f$. For this metric, a lower value means that fewer important features are incorrectly recognized as unimportant, which indicates higher faithfulness. However, one limitation of this formulation is that the appropriate values of the retention proportion $s$ can sometimes be unclear. While some tasks present natural values of $s$ to evaluate the metric with, this might not be the case for other tasks. Hase et al. (2021) propose averaging over different values of $s$ to avoid having to make an arbitrary choice.

Wang et al. (2020) extend to attribution methods that give real-valued (possibly negative) attribution scores to features. However, when quantifying faithfulness, they restrict to the ranking of only the $K^+$ features with positive attributions. Like Equation 10, they check if retaining the most important features are sufficient for computing the value(s) of the black-box $f$. The metric **sufficiency-ordering** is then formalized as follows:

$$\text{sufficiency-ordering}(E, f, \mathbf{x}) \triangleq \frac{1}{K^+ + 1} \sum_{s \in \{\frac{j}{K} | j = 0 \dots K^+\}} \min\{f(\mathbf{x}_{E_s}), f(\mathbf{x}_{E_{(\frac{K^+}{K})}})\} - f(\mathbf{x}_0) \quad (11)$$

In contrast to Equation 10, higher values mean fewer important features being recognized as unimportant, which indicates more faithfulness. Note that both formulations capture faithfulness as the sufficiency of the most important features in computing the model. The difference is that in Equation 10, higher sufficiency is represented by a smaller difference between the original model output and the output from retaining only the most important features, while in Equation 11, higher faithfulness is given by a larger difference between the the output from retaining only the most important features and the baseline output. Equation 11 is also different in that it clips scores to ensure that a few important features do not impact model output more than the set of all positive attributions. It also averages the impact of adding features to the baseline over all possible values of the retention proportion $s$. This allows for the user to not have to choose a specific retention proportion, which is also proposed by Hase et al. (2021).

The notion of *no false negatives* can also be generalized to evaluate concept bottleneck models, which predict higher-level concepts from features and then use these concepts to predict the target. Yeh et al. (2019b) illustrate that for classification tasks, the smaller the difference between the accuracy obtained by using just concept scores and the accuracy obtained by using the original input features, the more sufficient the set of concepts will be. Specifically, denote the inputs as $\mathbf{x}$ and the corresponding ground truth labels as $y$ in the validation set $\mathcal{V}$. Let $h$ be a mapping from the concepts to the prediction and $a_r$ be the random prediction accuracy that lower-bounds the metric score to 0. Then **concept-faithfulness** can be formalized as follows:

$$\text{concept-faithfulness}(E, f, \mathcal{V}) \triangleq \frac{\sup_h \mathbb{E}_{\mathbf{x}, y \in \mathcal{V}}[y = h(E(f, \mathbf{x}))] - a_r}{\mathbb{E}_{\mathbf{x}, y \in \mathcal{V}}[y = f(\mathbf{x})] - a_r} \quad (12)$$

Higher values mean that fewer important concepts are not captured, which indicates more faithfulness. Note that although this metric was originally termed as 'completeness', it evaluates the degree of *no false negatives* and hence differs from the more fitting notion of completeness discussed by Shrikumar et al. (2017) and Sundararajan et al. (2017), which measures both *no false negatives* and *no false positives* at the same time.

**Adversarial Attacks:** It is evident that having fewer false negatives is a strong indicator of faithfulness in explanations. However, adversarial attacks can be designed to compromise this property. Downstream tasks that have a strict requirement of ensuring no false negatives are likely to see undesirable consequences. An example of such a task can be an explanation under attack recognizing sensitive attributes as insignificant even though the underlying unfair model in fact depends on them. Slack et al. (2019) demonstrate that adversarial attacks can render post-hoc explanations such as LIME and SHAP **unfaithful** when explaining unfair models. In other words, LIME can be attacked such that it shows a higher degree of *false negatives*. This indicates that faithfulness in local post-hoc explanations is heavily dependent on input perturbations and this can be exploited by generating perturbed inputs from a different distribution. Dimanov et al. (2020) also showcase attacks that render LIME and SHAP unfaithful in the context of model fairness.

### 2.2.3 Faithfulness via No False Positives

Faithfulness as *no false positives* evaluates how well an explanation identifies truly insignificant features as insignificant. Again, this notion of fidelity is most applicable to and commonly used in feature attribution methods as well as certain exemplar methods.

In complement to (in)sufficiency (Equation 10), DeYoung et al. (2019) define the notion of **comprehensiveness** to measure faithfulness of feature attribution explanations that assign non-negative

importance values, by discarding the most important features that should impact the result significantly. Analogously, denote $\mathbf{x}_{E/s}$ as the input $\mathbf{x}$ where the proportion $s$ of the most important features, is discarded:

$$\mathbf{x}_{E/s}^{(k)} = \mathbf{x}_0^{(k)} \text{ if } k \in \{\text{Rank}_E(f,\mathbf{x})_1 \ldots \text{Rank}_E(f,\mathbf{x})_{\lceil sK \rceil}\} \text{ else } \mathbf{x}^{(k)} \tag{13}$$

Then, comprehensiveness is defined as:

$$\text{comprehensiveness}(f,\mathbf{x},E,s) \triangleq |f(\mathbf{x}) - f(\mathbf{x}_{E/s})| \tag{14}$$

For this metric, a higher value means that fewer unimportant features are incorrectly recognized as important, which indicates higher faithfulness. Like earlier, we present a generalized version using the absolute value and the averaging technique by Hase et al. (2021) can be used to avoid choosing an arbitrary $s$.

In contrast to sufficiency-ordering which retains the most important features (Equation 11), Wang et al. (2020) also propose measuring faithfulness in real-valued attributions by discarding the most important features. The impact on the output will then quantify how necessary those features are. As in Equation 11, they restrict to using the ranking of only $K^+$ features with positive attributions. The metric **necessity-ordering** can be defined as:

$$\text{necessity-ordering}(E,f,\mathbf{x}) \triangleq \frac{1}{K^+ + 1} \sum_{s \in \{\frac{j}{K} | j=0 \ldots K^+\}} \max\{f(\mathbf{x}_{E/s}) - f(\mathbf{x}_0), 0\}. \tag{15}$$

In contrast to Equation 14, lower values indicate fewer unimportant features being recognized as important, which indicates more faithfulness. As argued in the case of (in)sufficiency and sufficiency-ordering, the ideas behind comprehensiveness and necessity-ordering are similar. In Equation 14, the impact of discarding features is indicated by a larger difference between the original output and the output obtained by discarding (reverting) only the most important features. In Equation 15, the impact of discarding features is larger when we get a smaller difference between the output from reverting the most important features and the baseline output. Equation 15 is also different in that it clips scores to be non-negative, and averages the impact of discarding features over all possible values of $s$, as proposed by Hase et al. (2021).

Hsieh et al. (2020) similarly relate significance of feature to the robustness of the model explained when the feature is perturbed. However, instead of measuring the change in model output when a subset of features is perturbed to baseline, they go with the reverse direction and measure the minimum perturbation needed in a subset of features to change the model classification. Specifically, **robustness-S** if formalized as follows:

$$\begin{aligned}\text{robustness-S}(E,f,\mathbf{x},s) &\triangleq \\ \min\{\|\boldsymbol{r}\| \mid f(\mathbf{x}+\boldsymbol{r}) &\neq f(\mathbf{x}), \ \boldsymbol{r}_i = 0 \text{ for } i \notin \{\text{Rank}_E(f,\mathbf{x})_1 \ldots \text{Rank}_E(f,\mathbf{x})_{\lceil sK \rceil}\}\},\end{aligned} \tag{16}$$

where $f(\mathbf{x})$ and $f(\mathbf{x}+\boldsymbol{r})$ are the classifications of the model $f$ at the original input $\mathbf{x}$ and the perturbed input $\mathbf{x}+\boldsymbol{r}$ respectively, and only the top $\lceil sK \rceil$ feature are perturbed. Based on the idea that a more important subset $S$ should make the model less robust in its original prediction when reverted, it should correspond to a lower robustness-S value. Hence, they use **AUC under the curve**, where the x-axis is the subset size $\lceil sK \rceil = 1 \ldots K$ and the y-axix is the value of robustness-S at different revert proportion $s$, to evaluate faithfulness. A smaller AUC means a higher degree of *no false positives*, and is equivalent to averaging the impact of discarding features as in Equation 15.

While the above metrics measure faithfulness of attribution methods in general, Lundberg and Lee (2017) discuss faithfulness of additive feature attribution explanations in particular, where such an explanation is a linear function of an simplified input $\mathbf{x}^{(\text{simplified})} \in \{0,1\}^M$ from the original input $\mathbf{x} \in \mathbb{R}^K$. Formally, $E(f,\mathbf{x})(\mathbf{x}) = \phi_0 + \mathbf{x}^{(\text{simplified})T} E(f,\mathbf{x}) = \phi_0 + \sum_{i=1}^M \mathbf{x}_i^{(\text{simplified})} \cdot E(f,\mathbf{x})_i$, where $E(f,\mathbf{x})_i$ is the coefficient and also the attribution score for the $i$-th feature in the simplified input. They present **missingness** as an axiom of faithfulness to satisfy, requiring:

$$\mathbf{x}_i^{\text{simplified}} = 0 \quad \Rightarrow \quad E(f, \mathbf{x})_i = 0 \qquad \qquad \triangleright \text{Property 2 in Lundberg and Lee} \quad (17)$$

The implication means that in a linear representation, a value of $0$ has no impact to the output, so it should be attributed $0$ effect such that there is no *no false positive*. .

Besides feature attribution, the notion of *no false positives* also works for example-based methods. Let $E(f, \mathbf{x})$ be a set of examples selected by the explanation as the most responsible ones for the prediction $f(\mathbf{x})$, and $E(f, \mathbf{x})_i$ be the $i^{th}$ example in this set. Nguyen and Martínez (2020) then introduce **non-representativeness** to capture infidelity:

$$\text{non-representativeness}(E, f, \mathbf{x}) \triangleq \frac{\sum_i \mathcal{L}(f(\mathbf{x}), f(E(f, \mathbf{x})_i))}{|E(f, \mathbf{x})|} \qquad (18)$$

A lower value means that the model treat exemplars (examples) and the target observation $\mathbf{x}$ more similarly, which indicates that the selected examples are more representative and the explanation is more faithful.

## 3 Discussion and Conclusion

**Relationships and Trade-offs between Explanation Properties** Explanation properties serve as a proxy for the "goodness" of an explanation. The synthesis of the literature allows for a clear view of the relationships between explanation properties and trade-offs between these properties. Literature has indicated that preserving all properties for all explanations is not feasible and that the choice of an explanation and the properties it preserves, are tied to the task at hand. Understanding these relationships and trade-offs will then allow a user to prioritize properties correctly and tailor their choice to the task. We present an example of trade-offs below, other examples are in our full paper.

**Relationship between Faithfulness & Sensitivity** A good explanation should ideally be sufficiently faithful to the model while being minimally sensitive to input perturbations. Minimizing sensitivity naively would yield a constant and trivial explanation. Yeh et al. (2019a) discuss striking this balance by showing that explanation smoothing results in a simultaneous lowering of sensitivity and increase in faithfulness. This is because if explanation sensitivity is much larger than model sensitivity, the explanation infidelity will be lower bounded by the difference in explanation and model sensitivity. Lowering explanation sensitivity thus systematically helps lower infidelity. Another technique that can improve sensitivity and infidelity is adversarial retraining of the model.

**Faithfulness as Proportionality** Many explanations care about the contribution of each feature towards the model's output. Feature attributions attempt to do this directly by establishing a quantitative relationship between each feature and the model output. This has led to the emergence of proportionalities between the feature attribution explanation and the model output, which we discuss below. Methods vary primarily with regard to picking the relevant subset of features and comparing various subsets to the model output change.

Faithfulness in feature attributions requires the attributions to be proportional to the change in model output if those features were removed. This captures the true feature contributions towards the model output. Let $E(f, \mathbf{x})_i$ be the $i^{th}$ feature's attribution score, $\S$ be the set of important features and $\S^c$ be the complement of $\S$. Let $\mathbf{x}_{\S^c}$ be the input $\mathbf{x}$ with features in $\S^c$ retained and features in $\S$ reset to the reference value $\mathbf{x}_0$, i.e. $\mathbf{x}_{\S^c}^{(k)} = \mathbf{x}_0^{(k)}$ if $k \in \S$ else $\mathbf{x}^{(k)}$ for $k \in \S^c$.

$$\sum_{i \in \S} E(f, \mathbf{x})_i \propto \mathcal{L}(f(\mathbf{x}), f(\mathbf{x}_{\S^c})) \qquad (19)$$

Depending on the range of values that the attribution explanation can take, the loss term can assume different forms. For an explanation $E \in \mathbb{R}^K$ that includes negative attribution values, one possible loss $\mathcal{L}$ could be a simple difference:

$$\sum_{i \in \S} E(f, \mathbf{x})_i \propto f(\mathbf{x}) - f(\mathbf{x}_{\S^c}) \qquad (20)$$

When an explanation $E \in \mathbb{R}_{\geq 0}^{K}$ is restricted to non-negative attribution values, the loss $\mathcal{L}$ can also be the absolute difference:

$$\sum_{i \in \S} E(f, \mathbf{x})_i \propto |f(\mathbf{x}) - f(\mathbf{x}_{\S^c})| \tag{21}$$

This notion of proportionality captures both *no false positives* and *no false negatives*, and it can be decomposed to either one. For example, DeYoung et al. (2019) simplify Equation 21 to revert the most important features, which is proportional to a large change in model output. This is equivalent to a high **comprehensiveness** 14 value and a higher degree of *no false positives*. Similarly, retaining the most important features and reverting the other features should correspond to a small change in model output, i.e. a low **(in)sufficiency** 10 value and hence a higher degree of *no false negatives*.

**Conclusion** In this work, we have synthesized the ways in which two common explanation properties—sensitivity and faithfulness—have been formalized in the literature and described the strengths and weaknesses of each formalization. For a similar discussion of all the properties in Figure 2 and the trade-offs and relationships between them, see our full-length paper at `https://arxiv.org/abs/2211.05667`.

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
