# OpenReview forum: "What Makes a Good Explanation?: A Harmonized View of Properties of Explanations"
_NeurIPS.cc/2022/Workshop/TSRML — TSRML2022_

### Official Review · Reviewer_L6we · 2022-10-13
**A Survey on the Definitions of Sensitivity and Faithfulness in Explanation Methods**

**Overall Rating:** 5

**Summary:**

This paper surveys the definitions of sensitivity and faithfulness properties for post hoc explanation techniques. The discussion focuses on unifying prior literature in a single taxonomy and providing concrete mathematical definitions.

**Strengths:**

* The paper treats a question highly relevant to interpretable and explainable ML: what properties are essential for interoperability and explanation techniques, and how to evaluate them quantitatively?

* The paper includes concrete definitions for the described properties. It can be a handy technical reference for those looking for evaluation metrics for explanations.

* The discussion of trade-offs between properties is interesting; it would have made the paper more insightful to the reader acquainted with the topic if more of such trade-offs with concrete examples were described.

**Weaknesses:**

* As far as I understood, this review primarily focuses on the overview of the existing literature and does not present a new perspective or opinion. Moreover, there is no overview of related surveys with similar scopes, such as by Mohseni et al. (2021) and Nauta et al. (2022). These should be at least mentioned. Ideally, the novelty compared to the existing surveys on XAI evaluation should be emphasised.

* There is no formal justification for the selection of properties and references. Why were the particular papers chosen? Why does the review focus only on sensitivity and faithfulness?

* The review is blurry regarding the delineation between *ante hoc* and *post hoc* techniques. It might be beneficial to describe more clearly what method families are covered.

* It might be beneficial to add a table with the notation (rather than describe it in an unstructured text). That way, it will be easy for the reader to quickly refer to the notation guide (since the notation here is quite different from the literature). It is a matter of taste, but I would also avoid using $E(f)$ due to potential confusion with the expectation.

* The paper violates the recommended page limit (6 pages) by a margin. It seems that it is a shortened version of another manuscript which covers several other categories of explanation method properties. In the context of the current workshop, it would be helpful to shorten the paper and focus more explicitly on faithfulness and sensitivity. Otherwise, the paper does not read like an independent contribution.

### References

Mohseni, S., Zarei, N., & Ragan, E. D. (2021). A multidisciplinary survey and framework for design and evaluation of explainable AI systems. *ACM Transactions on Interactive Intelligent Systems (TiiS), 11*(3-4), 1-45.

Nauta, M., Trienes, J., Pathak, S., Nguyen, E., Peters, M., Schmitt, Y., ... & Seifert, C. (2022). From anecdotal evidence to quantitative evaluation methods: A systematic review on evaluating explainable AI. *arXiv:2201.08164*.

**Overall Recommendation:**

This paper surveys the concrete definitions of the sensitivity and faithfulness properties in explanation techniques. I lean towards rejection. While the problem of evaluating interpretable and explainable ML methods is an active research area, and this survey provides a helpful technical roadmap for some properties, the novelty of this work over other related surveys should be articulated more clearly. In particular, it is unclear if this survey brings a new perspective or identifies new relationships among the properties and their definitions.

**Review Confidence:**

3: The reviewer is fairly confident that the evaluation is correct

---

### Official Review · Reviewer_DFZx · 2022-10-18
**A summary of measures of evaluating local explanations**

**Overall Rating:** 6

**Summary:**

A summary and taxonomy of different robustness measures are presented in this study. The study includes a wide range of measures, such as faithfulness, sensitivity, and robustness.

**Strengths:**

- The studied problem,  i.e. the lack of standardized measures for local explanations, is a pressing issue in the explainability literature.
- The study includes a wide range of measures and does not limit itself to a small subset of measures.

**Weaknesses:**

- The authors do not provide a unification of these measures, instead they focus on summarizing different measures.
- For numerous measures presented in this paper, there is no analysis or discussion on the limitations and the strength of these measures.
- The paper is longer than eight pages

Some detailed comments:

Some of the related work is missing in the study:

* Evaluating the Quality of Machine Learning Explanations: A Survey on Methods and Metrics
* Evaluations and Methods for Explanation through Robustness Analysis: https://arxiv.org/abs/2006.00442


Some detailed comments:

> users expect explanations to be stable under minor changes to the input point x.

> In high-stakes applications, it is important to ensure that minor changes to the input do not lead to drastically different explanations since this could mislead users.

* My thought: What is the model the explanations explain is not stable? Do they still want the explanations to be stable?


* After Section 2.2.2, I had difficulty following your paper. For example, in Line 211, “retaining the most important feature should be sufficient for computing f.” What does computing f mean? In Line 226, your statement shows up again without further clarification.

* In Line 219, you have written: “the value of retention proportions can be unclear.” This applies to all robustness analysis methods, not insufficiency. See [1] for a discussion on this problem.

* In the section, Faithfulness as Approximation Accuracy, you have summarised numerous metrics that include a measure of change in model output, f(x), and explanations F(f(x)) however, your analysis is missing. My criticism of these measures is that they include two different concepts. One is a predicted probability point estimate, one is a flat vector of all features. In your review, you do not mention that comparisons such as this can lead to confusing results. So I ask you here: How can we evaluate explanations by measures that compare two measures with significantly different meanings?


[1]: Visualizing the Impact of Feature Attribution Baselines: https://distill.pub/2020/attribution-baselines/


**Overall Recommendation:**

I think that the paper is marginally above the acceptance threshold. The score can be improved if the authors can simplify the notations used in their study and include the limitations of these measures.

I hope the authors use the feedback from reviewers to improve the study.

**Review Confidence:**

3: The reviewer is fairly confident that the evaluation is correct

---

### Decision · Program_Chairs · 2022-10-23

**Decision:**

Accept

**Comment:**

This paper is a good survey paper. Please improve the paper based on reviewers' suggestions.